



# Parameterizing the vertical downward dispersion of ship exhaust gas in the near-field

Ronny Badeke[1], Volker Matthias[1], David Grawe[2]

[1]Institute of Coastal Research, Helmholtz-Zentrum Geesthacht, 21502 Geesthacht, Germany
[2]Center for Earth System Research and Sustainability (CEN); Meteorological Institute, Universität Hamburg, 20146 Hamburg, Germany

*Correspondence to*: Ronny Badeke (Ronny.Badeke@hzg.de)

**Abstract.** Estimating the impact of ship emissions on local air quality is a topic of high relevance, especially in large harbour cities. For chemistry transport modeling studies, the initial plume rise and dispersion play a crucial role for the distribution of

pollutants into vertical model layers. This study aims at parameterizing the vertical downward dispersion in the near-field of a prototype cruise ship, depending on several meteorological and technical input parameters. By using the micro-scale transport and stream model MITRAS, a parameterization scheme was developed to calculate the downward dispersion, i.e. the fraction of emissions, which will be dispersed below stack height. This represents the local concentration in the vicinity of the ship. Cases with and without considering the obstacle effect of the ship have been compared. Wind speed and ship size were found

to be the strongest factors influencing the downward dispersion, which can reach values up to 55 % at high wind speed and lateral wind. This compares to 31 % in the case where the obstacle effect was not considered and shows the importance of obstacle effects when assessing the ground-level pollution situation in ports.

## 1 Introduction

Ship emissions are among the harmful anthropogenic influences on air quality and human health, especially in big harbor cities. Regarding air quality, this covers various gaseous pollutants like sulfur dioxide ($SO_2$), oxides of nitrogen ($NO_x$), as well as particulate matter (PM).

Corbett et al. (2007) presented a study on the global effect of ship-related particulate matter emissions on human health and found that they are at least partially responsible for around 60,000 cardiopulmonary and lung cancer deaths annually, mainly

in coastal regions of Europe, South and East Asia. Despite numerous measures to reduce these impacts like the International Maritime Organization's global sulfur cap to 0.5 % maximum sulfur content in marine fuels since 1 January 2020 (IMO, 2020), research on the impacts of ship emissions on human health and air quality remains an ongoing topic (Barregard et al., 2019; Broome et al., 2016; Lin et al., 2018; Liu et al., 2016; Ramacher et al., 2019; Sofiev et al., 2018; Zhang et al., 2019).

The urge to quantify ship emission gained even more attention due to the fast growth in shipping activity during the last decades

(Brandt et al., 2013; United Nations Conference on Trade and Development, 2019). Many ship plume modeling studies focus on global or regional scale plume dispersion, chemistry and their parametrization (e.g. Aksoyoglu et al., 2016; Huszar et al.,



2010; Vinken et al., 2011). For emission calculation they can make use of global (Corbett et al., 2007; Wang et al., 2008) or regional (Aulinger et al., 2016; Jalkanen et al., 2009, 2012) shipping emission inventories. Increasing trends in ship emissions in Northern Europe have been modelled for the North Sea (Matthias et al., 2016) and Baltic Sea (Karl et al., 2019) and the

efficiency of emission reduction measures have been evaluated.

Matthias et al. (2018) pointed out the importance of correct spatial and temporal distribution of emissions in chemistry transport models, which are connected with uncertainties. The distribution depends strongly on data availability, interpolation procedures and initial assumptions. For example, an emission overestimation for single ships can occur, if ship emissions are diluted instantaneously and equally into a large grid (von Glasow et al., 2003; Vinken et al., 2011). This problem can be

overcome by using smaller scale models and a bottom-up approach for emission inventories (Aulinger et al., 2016; Eyring et al., 2010; Jalkanen et al., 2009, 2012; Johansson et al., 2017).

However, new problems arise when modeling is performed on a single-ship level. On the technical side, this includes the correct localization of ships, which can nowadays be done by using Automatic Identification Systems (AIS). The emission factors are calculated as a function of several technical parameters like fuel type, engine load, engine power, engine type, ship

size, ship class, as well as the performed activity (e.g. cruising, maneuvering, or berthing), the latter to be derived from AIS data (Aulinger et al., 2016).

But also new methods for spatially allocating the emissions from big ships need to be developed, since the emission height often lies significantly above ground level. The exhaust gas leaves the stack with a certain exit velocity and a temperature of several hundred °C. These quantities depend on technical parameters of the individual ship, which are often unknown.

Many analytical single plume models are based on Gaussian dispersion formulas (Briggs, 1982; Janicke and Janicke, 2001; Schatzmann, 1979). This means that the pollutant distribution corresponds to a normal probability distribution. An example is the Offshore and Coastal Dispersion (OCD) algorithm of Hanna et al. (1985). An accurate representation of plume rise and downward dispersion processes in the near-field under different meteorological conditions is important, since it changes the effective emission height and may cause the vertical concentration profile to deviate from a Gaussian shape (Bieser et al.,

2011; Brunner et al., 2019). Based on a Large Eddy Simulation study, Chosson et al. (2008) pointed out that Gaussian plume dispersion models might not be well suited for the early plume development.

Despite of not running at full engine power inside of the harbor, ocean-going ships still consume large amounts of fuel for heat and electricity production and therefore emit atmospheric pollutants while at berth (Hulskotte and Denier van der Gon, 2010). These have been found to be up to 5 times higher compared to other activities like maneuvering or cruising during the course

of a year, as ships spend more time at berth and also have a high auxiliary engine power demand for hotel services (Tzannatos, 2010). This can lead to severe air quality problems in harbor areas. Murena et al. (2018) applied a computational fluid dynamics model to assess the impact of cruise ship emissions on the facades of waterfront buildings in Naples, Italy. The highest $SO_2$ concentrations were modelled for areas nearby the ships and on the port front facades of the first line of buildings.

More research is needed to better understand the effect of technical and meteorological parameters on the downward dispersion

process that causes these strong pollution scenarios inside of a harbor. Although the OCD model of Hanna et al. (1985) includes



effects of pollutant downward dispersion behind the obstacle, i.e. the vessel, by lowering the effective plume height and adjusting dilution parameters in the model, this effect has yet to be applied to large ships.

The aim of this study is to quantify the main factors that have an impact on the downward dispersion process for a large cruise ship. For this purpose, an Eulerian micro-scale model is used to calculate the downward dispersion in close vicinity to the ship.

Furthermore, a parameterization is developed for the downward dispersion in dependence of the crucial meteorological influences and the technical specifications of the ship. Finally, it will be shown under which conditions the obstacle-effect on the downward dispersion needs to be considered.

## 2 Methodology

The dispersion of an exhaust plume is affected by several meteorological and technical parameters (Fig. 1). The upward

movement, i.e. the plume rise, is mainly determined by the initial temperature of the exhaust and its exit velocity, which can be calculated by dividing the gas volume flow by the stack diameter. The stack angle describes whether the exhaust flow is directed vertically, horizontally or at an angle. The stack height only has an indirect effect on the plume rise, as higher emitted gases experience a stronger wind speed inside the boundary layer.

Turbulence enhances the plume dispersion, leading to dilution of the embedded gases by entrainment of ambient air into the

plume. The dispersion increases with the wind speed. It depends also on the ship geometry and the flow direction of the wind towards the vessel. Furthermore, a stronger turbulence occurs in case of higher surface roughness.

The ambient vertical temperature profile determines the atmospheric stability. The presence of an inversion can strongly decrease the strength of the plume dispersion, as it thermodynamically hinders the vertical movement of air masses. Depending on the altitude of the inversion and the exhaust temperature, the plume may or may not break through the inversion.


## 2.1 MITRAS

The micro-scale transport and stream model (MITRAS) is a non-hydrostatic, three-dimensional Eulerian model, based on the Navier–Stokes-equations, the continuity equation and the conservation equations for scalar properties like temperature, humidity and trace gas concentrations (Grawe et al., 2013; Salim et al., 2018; Schlünzen et al., 2003, 2012). It accounts for

obstacle-induced turbulence on the wind field as well as effects of thermal stratification.

In this study, a non-equidistant grid is used with the highest resolution of 2 m x 2 m x 2 m close to the ship. The chosen domain has an overall size of around 1 km x 1 km horizontally and 500 m vertically. The surface cover for the whole domain is water and the roughness length is calculated from the wind speed (see Schlünzen et al., 2012, for detailed equations). Its values are close to zero.

The emission occurs continuously in one model cell right above the ship stack, which is an impenetrable obstacle cell (Fig. 2). The emitted gas is as a passive trace gas (e.g. $CO_2$ or non-reactive $SO_2$). No chemical reactions occur in the simulations. The





emission cell has a constant temperature, which corresponds to a given exhaust temperature and a vertically directed exhaust velocity. The wind field is affected by Coriolis force and friction force, which cause the wind to slightly turn counterclockwise according to an Ekman spiral. Furthermore, the flow field is modified by the obstacle itself, the high temperature of the exhaust

and the exit velocity. No deposition occurs in the model domain, the surface is a mirror source which reflects the concentration when the lowest model layer is reached.

## 2.2 Meteorological data

Idealized meteorological conditions are used to investigate effects of single variations of input parameters on the dispersion process. The range of input values is listed in Table 1. One input parameter per model run was varied while the other

meteorological and technical parameters were fixed at predefined default values.

The ambient temperature is set to 15 °C at the surface. It changes with altitude according to the given ambient temperature gradient, which represents the atmospheric stability. The value of ambient temperature itself has a negligible small effect on the plume dispersion compared to the plume temperature and was therefore not varied in this study.

The atmospheric stability is varied in a range of different lapse rates, covering one unstable condition (-1.2 K · 100 m$^{-1}$), one

neutral condition (-0.98 K · 100 m$^{-1}$) and several stable conditions including inversions (up to +0.5 K · 100 m$^{-1}$).

The wind speed is investigated in a range of 2–15 m s$^{-1}$. The limits were chosen according to hourly wind speed data from Hamburg weather mast in 2018 (see Appendix A) and can also be seen as representative for other large North Europe ports including Rotterdam and Antwerp. The value 2 m s$^{-1}$ is close to the 5$^{th}$ percentile and 15 m s$^{-1}$ corresponds to the 95$^{th}$ percentile at 280 m measurement height. This covers most of the naturally occurring scenarios. The selected default value is 5 m s$^{-1}$ which

fits well with the mean wind speed in Hamburg at a height of 50 m, which is close to the stack height.

The effect of wind direction is relevant in correspondence to the orientation of the ship. Frontal wind is herein defined at an angle of 0° and lateral wind at 90°. Oblique wind conditions lie between these values.

## 2.3 Ship characteristics

This study represents a cruise ship prototype. From an online database (Port of Hamburg, 2020) the average length and width

of cruise ships that were visiting Hamburg harbor during the years 2018 to 2019 has been calculated. The stack height was approximated from freely available photos (e.g. Vesseltracker, 2020). The ship prototype has a length of 246 m, a width of 30 m and a stack height of 52 m (see Table 1 and Fig. 3). This corresponds to a typical cruise ship that can carry between 1000 and 2500 passengers. A non-moving source is assumed, i.e. a hoteling ship at berth.

The study goes beyond a case study. A loaded container ship of similar size and exhaust characteristics would deliver similar

results because its shape is comparable. On top of that, for all investigated input characteristics, the results of stack-only cases are presented as well. Therefore, one can assume that results for smaller ships lie between these two cases.

The exhaust gas temperature depends on technical parameters of the ship's engine and can be found in engine data sheets provided by manufacturers like Caterpillar (CAT, 2020), Wärtsilä (Wärtsilä, 2020) and MAN (MAN, 2020) on their websites.





For large cruise vessels, it ranges between approximately 300 °C and 400 °C, depending on the used engine power. However,

the exhaust temperature can be lowered by 75–100 °C when a heat exchanger which generates electric energy from the excess heat is in operation (Murphy et al., 2009). Therefore, the temperature effects are investigated for 200 °C, 300 °C and 400 °C plumes to cover a realistic spectrum. Similarly, the exit velocity was assumed from these data sheets. It depends on the engine type (main engine or auxiliary engine) and the used engine power and was investigated in a range of 4–12 m s$^{-1}$.

### 2.4 Plume dispersion in different regimes

When investigating plume dispersion, one needs to separate two regimes: the momentum-driven regime and the buoyancy-driven regime. In the momentum-driven regime the movement of the plume is affected by (a) the initial plume rise due to both, the exit velocity and the high-temperature convective upward transport and (b) the dispersion due to turbulence generated by the obstacle (i.e. the ship) inside the wind field. In the buoyancy-driven regime, the movement of the plume is determined by the wind field and turbulence generated by the ambient conditions (e.g. orography effects and surface roughness). Here, the

plume temperature is equal to the ambient temperature. The micro-scale model MITRAS can investigate plume behavior in both regimes on a high resolution.

MITRAS is used to capture the initial plume rise and turbulence effects in the momentum driven regime. The vertical concentration profiles are calculated at a distance outside of the momentum-driven regime, i.e. when the buoyancy-driven regime is reached. Then, the concentration profiles are calculated on a 100 m x 100 m area column with layer-mean values

(Fig. 4). The calculation of these column values has two benefits. First, it covers the mean behavior of the whole plume better than single values of 2 m x 2 m x 2 m grid sizes, since the movement of the plume can be highly variable. Second, the concentration profiles can then also be transferred into larger-scale models which usually have a much coarser grid (e.g. 100 m x 100 m horizontally). However, the coupling of MITRAS results into a larger-scale model will be part of a future study and is not covered here.

Since the plume needs to have cooled down to ambient temperatures to be considered outside the momentum-driven regime, test simulations have been performed to find a distance at which this condition is met (see Appendix B). This was the case at a distance of 100 m downwind of the ship. Therefore, all concentration profiles are calculated as 100 m x 100 m columns with average concentration per layer at a distance of 100 m downwind of the ship.

In the following, the term **downward dispersion D** is defined as the relative proportion of the total concentration column in

the layers below the stack height.

$$D = \frac{\int_0^{h_{stack}} c}{\int_0^{h_{top}} c} \cdot 100 \ \% \ ,  \tag{1}$$

where $h_{top}$ is the altitude of the highest model layer (500 m), $h_{stack}$ is the stack height (52 m) and c is the total concentration. A mean downward dispersion is calculated for the described 100 m x 100 m column at a distance of 100 m downwind from the



stack. From an application perspective, this downward dispersion parameter is an indicator for the pollution situation in the
vicinity of the ship and useful to evaluate the level of pollution inside of a harbor.

For single regression analyses, downward dispersion values are investigated depending on the variation of one single input
parameter at a time while the others remain at default settings (Table 1). To assess the sensitivity of the downward dispersion
to each input parameter, an **effective range r** is calculated. It is defined as the difference between the highest and the lowest
downward dispersion value for one regression:

$$r_i = D(max)_i - D(min)_i,$$    (2)

where i is the individual input parameter that is varied while the other remain at default setting. The effective range describes
how strongly one parameter can change the downward dispersion and helps to evaluate which input parameter has the strongest
impact in the given range of values.

## 3 Results and discussion

Figure 5 presents an exemplary output of the MITRAS model for the default conditions, i.e. frontal wind at 5 m s$^{-1}$, exit velocity
of 10 m s$^{-1}$, exhaust temperature of 300 °C and an ambient temperature gradient of -0.65 K · 100 m$^{-1}$. The concentration values
result from an emission of 50 kg trace gas per hour.

The following subsections describe the results of single- and multi-parameter regressions that were performed in order to
describe the relationship between the downward dispersion and the input parameters. From the multi-parameter regression, a
parameterization is derived that covers all input parameters in the investigation range. A bootstrapping procedure is presented
to test how well the parameterization results match with the MITRAS model results. The obstacle effect is evaluated and,
finally, some limitations of the modeling approach are discussed.

### 3.1 Results of single-parameter regressions

Single-parameter regressions are performed after basic statistic formulae (see Appendix C) to investigate the impact of
individual input parameters, i.e. wind speed, exit velocity, wind direction, plume temperature and atmospheric stability on the
downward dispersion.

### 3.1.1 Effect of wind speed and exit velocity

The    dependence    of    the    downward    dispersion    from    wind    speed    was    modelled    in    the    range    of
2–15 m s$^{-1}$ at the uppermost model layer, which is set as the input parameter. It is slightly lower at stack height following the
logarithmic vertical wind profile. See Table C1 for the exact values at stack height.

Figure 6 presents results of a single linear regression for the dependence of downward dispersion on varying wind speeds with and without obstacle effect. Other input parameters remained constant at default values (Table 1). A linear relationship with correlation coefficients $R^2$ of 0.98 was found for both runs with and without ship, respectively. At high wind speeds, the turbulence behind the obstacle causes strong downward dispersion. Under these settings, the wind speed has an effective range on the downward dispersion of 40.3 % with and 21.1 % without ship, making the wind speed a crucial factor influencing the downward dispersion (Fig. 6 and Table 2).

A similarly strong linear relationship has been found between the exit velocity of the exhaust gas and the downward dispersion (Fig. 7 and Table 2) with regression $R^2$ of 1.00 for cases with and without obstacle. It is, however, a negative dependence, because higher exit velocities transport the plume into higher altitudes and consequently the downward dispersion is lower. The effective range is much smaller than for the wind speed with only 3.7 % with and 2.1 % without obstacle, respectively.

### 3.1.2 Effect of wind direction

The strength of the downward dispersion was investigated depending on different wind directions in relation to the orientation of the ship. Frontal wind (angle 0°) hits the short side of the vessel, which has a width of 30 m, whereas lateral wind (angle 90°) has to be lifted over the 246 m length of the ship. Therefore, a stronger distortion of the flow during lateral wind has been observed.

The downward dispersion correlates linearly with the cosine of the flow angle φ (Fig. 8). A regression coefficient $R^2$ of 0.98 was calculated. At default settings a downward dispersion ratio of 7.0 % and 16.6 % was found under frontal and lateral wind conditions, respectively. This results in an effective range of 9.6 %. The corresponding downward dispersion under no obstacle condition is 2.3 %. There is no effective range for no-obstacle conditions, because here a single symmetrical stack is assumed, where the downward dispersion values are the same for both, frontal and lateral wind. However, very small differences between these conditions can occur during the modeling (see Table C1), which result from an asymmetry in the numerical grid.

### 3.1.3 Effect of exhaust plume temperature

The exhaust plume temperature depends on technical parameters like the engine power and the use of a heat exchanger and, therefore, a range of possible temperatures (200 °C–400 °C) was investigated. Figure 9 presents results of the single linear regression for the downward dispersion at varying exhaust temperatures with and without obstacle effect.

Once again, a strong linear relationship with correlation coefficients $R^2$ of 0.98 and 0.99 was found for results with and without ship, respectively. At higher exhaust temperatures, the plume reaches higher altitudes by convective upward movement, which results in lower downward dispersion ratios. The effective range under default settings is 6.9 % with and 2.9 % without obstacle effect (Table 2).



### 3.1.4 Effect of atmospheric stability

The effect of atmospheric stability $\Gamma$ on the downward dispersion was investigated in a range from unstable (-1.2 K $\cdot$ 100 m$^{-1}$) to very stable (+0.5 K $\cdot$ 100 m$^{-1}$) vertical temperature gradients. Under default settings, linear regression resulted in correlation coefficients of $R^2 = 0.90$ and 0.94 with and without ship, respectively (Fig. 10a). Since the $R^2$ coefficient was low compared

to the other investigated input parameters, a linear dependence would deliver poorer results for this parameter. Therefore, a quadratic dependence was calculated as well.

Since the square of a negative vertical temperature gradient would result in a positive value, a sign function was applied. The mathematical expression is:

$$\text{sgn}(x) := \begin{cases} -1 \ if \ x < 0 \\ 0 \ \ if \ x = 0 \\ 1 \ \ if \ x > 0 \end{cases} \qquad (3)$$

Then, the correlation between downward dispersion and $\text{sgn}(\Gamma)\Gamma^2$ is calculated (Fig. 10b). It shows better agreement in the cases considering obstacle effects ($R^2 = 0.99$) and slightly better agreement in cases without ship ($R^2 = 0.96$), as well. It is a negative correlation, because higher temperature gradients correspond to a higher stability which thermodynamically prevents the plume to disperse vertically and therefore lowers the downward dispersion ratio. The effective range of the temperature gradient on the downward dispersion is 6.6 % for ship cases and 3.8 % for stack-only cases.

## 3.2 Result of the multiple regression

Multiple regression is performed according to the equations in Appendix C2. The downward dispersion ratio depends linearly on all investigated input parameters, their cosine (in case of the angle of wind direction) or their squares (in case of atmospheric stability). With that in mind, a training data set for the multiple regression was created. Here, all independent input parameters are varied at the same time (but in the given range) and the downward dispersion ratio is calculated with MITRAS. For a set

of 39 different combinations (Table C1) of input parameters with obstacle effect and 27 without, the estimation coefficients $\hat{\beta}_i$ for individual parameters i (wind speed, exit velocity, etc.) are calculated with the multiple regression. The number of simulated cases without obstacle effects are lower, because in these cases the wind direction has been varied which will not show differences in case of stack-only conditions. The resulting formulae for the parameterization read

$$D \ [\%] = 13.03 + 3.45 \ v_{wind} - 1.01 \ v_{exit} - 0.026 \ T_{exh} - 3.81 \ \text{sgn}(\Gamma)\Gamma^2 + 6.13 \cos(\phi), \qquad (4)$$

with ship and

$$D \ [\%] = 4.55 + 1.78 \ v_{wind} - 0.64 \ v_{exit} - 0.018 \ T_{exh} - 3.40 \text{sgn}(\Gamma)\Gamma^2, \qquad (5)$$

without ship (i.e. stack-only).

Here, $v_{wind}$ and $v_{exh}$ are given in [m s$^{-1}$], $T_{exh}$ in [°C], $\Gamma$ in [K / 100 m] and $\varphi$ in [°], where 0° refers to frontal wind and 90° to lateral wind.



### 3.3 Bootstrapping

A bootstrapping procedure is applied to estimate how well the parameterization can represent the model data. For this purpose, downward dispersion ratios were calculated with the parameterization formulae (Eq. 4 and 5) and compared to the original MITRAS results for all investigated cases and ranges. Results of the individual parameterization results are listed in Table C1 and Table 3 gives the overall results of the bootstrapping procedure.

With a mean absolute error of 1.9 ± 1.6 % for cases with ship and 1.2 % ± 0.9 % without ship the parameterization delivers very similar results to the model runs. The maximum absolute errors were found to be 6.1 % in cases with ship and 4.0 % in cases without ship.

### 3.4 Assessment of the obstacle effect

Another aim was to investigate under which conditions the strongest downward dispersion occurs and which effect the consideration of the obstacle has on the downward dispersion.

From the single-parameter regressions, it is assumed that the strongest downward dispersion occurs at high wind speed (15 m s$^{-1}$) with lateral wind (90°), low exit velocity (4 m s$^{-1}$), low plume temperature (200 °C) and during unstable atmospheric conditions (-1.2 K · 100 m$^{-1}$). This is displayed in Fig. 11 with the ship as an obstacle (panel a) and under stack-only conditions (panel b).

The calculated downward dispersion ratio for this condition is 54.9 % and 31.1 % with and without obstacle effect, respectively. This means that a significant proportion of nearly 25 % of the emission can be dispersed downwards only by taking into account the turbulence caused by the ship.

### 3.5 Discussion of limitations

Despite efforts to represent real conditions as best as possible, the results are subject to a few limitations or uncertainties that will be discussed in the following section.

One factor that is not considered in this study is relative humidity. Here, a distinction must be made between the relative humidity of the ambient air and the relative humidity of the exhaust. By using the Lagrangian concept based on the so-called projected area entrainment (Lee and Cheung, 1990), Affad et al. (2006) stated that the relative humidity of the ambient air has only a slight impact on the plume rise, diameter and temperature for values between 20 and 90 %. It can have an impact on particle growth, but as this study focusses on a passive gaseous tracer, this effect is neglected. On the other hand, the humidity of the exhaust gas might have a larger impact on the plume rise. Since water vapor has a lower density than air, an exhaust gas mixture of high humidity will show a stronger plume rise. Furthermore, as the gas will quickly condense, it will release latent heat and rise further. However, the data base on humidity of ship exhaust is sparse. It could play a role in case of vessels using a scrubber to wash out $SO_2$ from the exhaust. During this process, the exhaust is cooled down significantly and therefore will





show a weaker plume rise (Murphy et al., 2009). It is unclear, if the additional buoyancy can compensate for the lower exhaust temperatures. Due to these uncertainties and lack of data, the relative humidity has not been included in this study.

Second, the emission is assumed to occur in the grid cell above the stack, which has a size of 2 m x 2 m x 2 m. This corresponds to a stack with a square cross section of 4 m² and is a limitation connected to the chosen grid size. Real stacks are usually round and have a smaller diameter. The real exit velocity could therefore differ slightly. However, by comparing the effective

ranges for exit velocity against all other input factors (Table 2), it can be seen that this parameter has the smallest overall impact of the downward dispersion and therefore, this uncertainty factor has a low impact on the overall performance.

Another assumption was that the ship has been considered as a non-moving source, i.e. a hoteling ship. However, the results can be applied to a moving ship by calculating the vector sum of the wind and the vessel speed. It is difficult to account for complex maneuvers, though, as the resulting wind vector may change quickly and the technical conditions like exhaust

temperature and exit velocity may also vary with the speed of the ship.

The chosen model surface is water but assuming a hoteling ship, the land surface effects may play a role for the dispersion. This effect has not been part of this study, as this is a highly variable parameter that depends on the structure of the harbor, the city and the orography. These effects need to be covered by a larger scale model.

## 4 Conclusion

A ship plume modelling study was performed with the micro-scale numerical model MITRAS to investigate the downward dispersion of the exhaust in close vicinity to a modelled cruise ship (i.e. in the momentum-driven regime). A set of 39 different scenarios with varying meteorological and technical input parameters were analyzed. A multiple regression algorithm was used to estimate a parameterization function for the downward dispersion. This parameterization has been tested against the MITRAS model results through a bootstrapping procedure.

From single-parameter regressions a positive linear relationship of the downward dispersion from wind speed and negative linear relationships from exit velocity, plume temperature and the cosine of the angle of wind direction was found. The downward dispersion ratio was larger in case of lateral wind than in case of frontal wind. In case of atmospheric stability, the downward dispersion showed a squared dependence from the vertical temperature gradient multiplied by the sign function. From all these input parameters, the wind speed shows the largest effect on the downward dispersion in the investigated range

$(2\text{–}15 \text{ m s}^{-1})$.

A comparison of the model results and the parameterization from multiple regression shows a good agreement with a mean absolute error of $1.9 \pm 1.6$ % for cases with ship and $1.2 \pm 0.9$ % without ship. For the case of strongest downward dispersion, the difference was calculated between downward dispersion with (54.9 %) and without considering the obstacle effect (31.1 %), which was almost 25 %. This shows how important it is to consider the effects of the downward dispersion in the

momentum-driven regime when one wants to evaluate the air pollution situation in harbor areas.



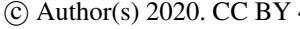

The parameterization functions can also be used for container ships of similar size. It may also be applied to different emission situations like industrial stacks.

In a future study, other plume parameters will be derived from the vertical concentration profiles in a similar way as the downward dispersion. This includes the height of the plume axis and the shape of the vertical plume profile, which may deviate

from the often assumed Gaussian distribution. These results can further be used in a city-scale model, which only calculates the plume dispersion inside the buoyancy-driven regime.

**Author contribution**

Conceptualization: all authors; methodology: R. Badeke and D. Grawe; calculation of results: R. Badeke; discussion and conclusion: all authors; writing: R. Badeke. All authors have read and agreed to the published version of the manuscript.

**Code/Data availability**

All regression results can be obtained by applying the functions in Appendix C on the data of Table C1. A data table "regression_data.csv" and a Python script "multiple_regression.py" have been added as supplementary material.

**Conflict of interest**

The authors declare no conflict of interest. The funders had no role in the design of the study; in the collection, analyses, or

interpretation of data; in the writing of the manuscript, or in the decision to publish the results.

**Acknowledgment**

This work was funded by the German Science Foundation (DFG) in the framework of DFG-NSFC funded project ShipChem. The authors would kindly like to thank Prof. Dr. Heinke Schlünzen, Prof. Dr. Bernd Leitl and Prof. Dr. Kay-Christian Emeis for the fruitful discussions during the preparation of this manuscript. We would further like to thank the Meteorological

Institute of University Hamburg for providing wind data of the Hamburg weather mast.

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

## Appendix

### A: Wind speed in Hamburg

Input data for wind speed data were used from Hamburg weather mast, provided by the Integrated Climate Data Center (ICDC)

(ICDC, 2020). The weather mast is positioned at a meteorological measurement station in Billwerder, Hamburg (53° 31′ 09.0″ N, 10° 06′ 10.3″ E). Hourly data from the full year 2018 were statistically analyzed at five different measurement heights (Fig. A1, Table A1).

### B: Plume temperature

Figure B1 presents results for maximum temperatures in the MITRAS domain for one case with the highest temperature (400

°C). Ambient temperatures (15 °C) are reached at a horizontal distance of approximately 100 m from the stack.

### C: Regressions

This section describes the general application of linear and multiple regression on the model results.

### C.1: Single linear regression

A simple approach to estimate a target variable Y (e.g. the downward dispersion) from one single independent variable X (e.g.

the wind speed) is a linear regression in the form of

$$\hat{Y} = \widehat{\beta_0} + \widehat{\beta_1} X \,, \tag{C1}$$

where $\beta_0$ and $\beta_1$ are the ordinate axis intersection and the slope, respectively, and the circumflex (^) describes an estimated parameter. $\widehat{\beta_0}$ and $\widehat{\beta_1}$ are calculated with the least squares method, minimizing the quadratic deviation between model result values $Y_i$ and estimated values $\hat{Y}_i$. The required function Q reads:





$\quad Q(\widehat{\beta_0}, \widehat{\beta_1}) = \sum_{i=1}^{n}(Y_i - \widehat{Y_i})^2 = \sum_{i=1}^{n}(Y_i - \widehat{\beta_0} - \widehat{\beta_1}X_i)^2$ $\qquad$ (C2)

Minimizing is done by applying partial derivations from Q to $\widehat{\beta_0}$ and $\widehat{\beta_1}$. This results in

$\widehat{\beta_0} = \bar{Y} - \widehat{\beta_1}\bar{X}$, $\qquad$ (C3)

$\widehat{\beta_1} = \frac{\sum_{i=1}^{n}\left((X_i - \bar{X})(Y_i - \bar{Y})\right)}{\sum_{i=1}^{n}(X_i - \bar{X})^2}$, $\qquad$ (C4)

where $\bar{X}$ and $\bar{Y}$ are the mean values of the corresponding dataset.

**C.2: Multiple regression**

The variable Y can depend on more than one independent input variable ($X_1$, $X_2$, …, $X_p$). Then, a multiple regression can be applied and in the case of linear dependencies, the corresponding regression is called multilinear. The multilinear estimation for $\hat{Y}$ reads:

$\hat{Y} = \widehat{\beta_0} + \widehat{\beta_1}X_1 + \widehat{\beta_2}X_2 + \dots + \widehat{\beta_p}X_p$ $\qquad$ (C5)

$\quad$ Again, the minimum distance between $Y_i$ and $\widehat{Y_i}$ can be calculated by the least squares method, similar to the case of linear regression, by minimizing the function Q:

$Q(\widehat{\beta_0}, \widehat{\beta_1}, \widehat{\beta_2}, \dots \widehat{\beta_p}) = \sum_{i=1}^{n}(Y_i - \widehat{\beta_0} - \widehat{\beta_1}X_{i,1} - \widehat{\beta_2}X_{i,2} - \dots - \widehat{\beta_p}X_{i,p})^2$ $\qquad$ (C6)

However, as this can lead to complicated expressions of $\hat{\beta}$, one can make use of a matrix representation.

$$\begin{pmatrix} \widehat{Y_1} \\ \widehat{Y_2} \\ \vdots \\ \widehat{Y_n} \end{pmatrix} = \begin{pmatrix} 1 & X_{11} & X_{12} & \cdots & X_{1p} \\ 1 & X_{21} & X_{22} & \cdots & X_{2p} \\ \vdots & \vdots & \vdots & \ddots & \vdots \\ 1 & X_{n1} & X_{n2} & \cdots & X_{np} \end{pmatrix} \cdot \begin{pmatrix} \widehat{\beta_0} \\ \widehat{\beta_1} \\ \vdots \\ \widehat{\beta_p} \end{pmatrix} \qquad (C7)$$

$\quad$ By using both the transpose ($^T$) and the invert ($^{-1}$) operator, the equations can be transformed to a general solution for $\hat{\beta}$:

$$\begin{pmatrix} \widehat{\beta_0} \\ \widehat{\beta_1} \\ \vdots \\ \widehat{\beta_p} \end{pmatrix} = (\boldsymbol{X}^T\boldsymbol{X})^{-1} \cdot \boldsymbol{X}^T \cdot \begin{pmatrix} \widehat{Y_1} \\ \widehat{Y_2} \\ \vdots \\ \widehat{Y_n} \end{pmatrix} \qquad (C8)$$

Table C1 presents an overview on the results of the multiple regression.





**Figures**

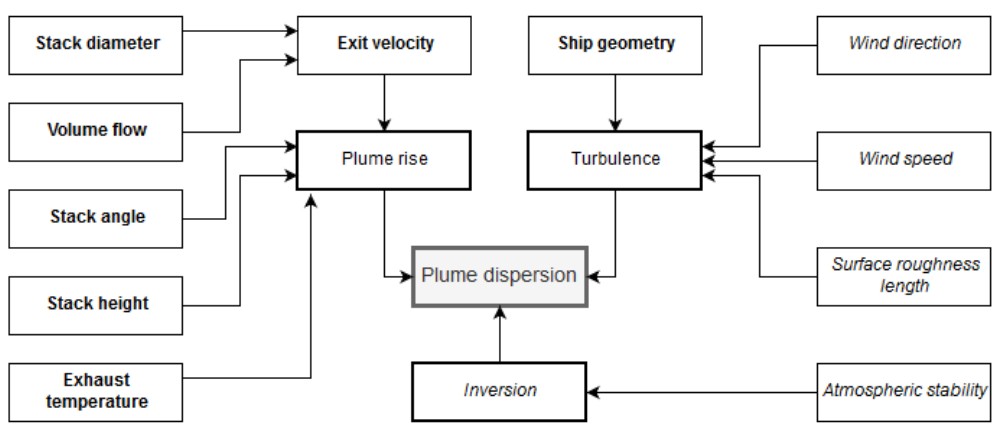


**Figure 1: Conceptual model of parameters affecting the shape and movement of a ship plume (bold: technical parameters, italic: ambient parameters)**

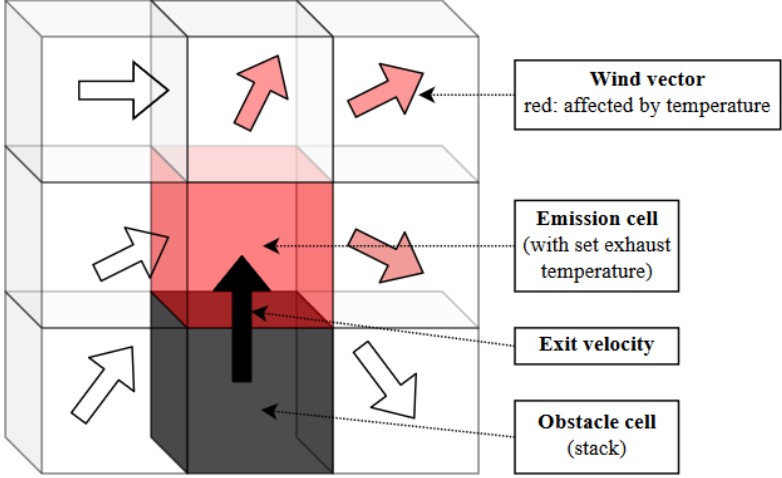

**Figure 2: Visualization of the stack emission for wind direction from left to right. Passive trace gas emission occurs in the cell above the stack, which has a constant exhaust temperature and a vertically directed exhaust velocity. The arrows indicate the change of the ambient wind field due to the obstacle and the plume temperature.**






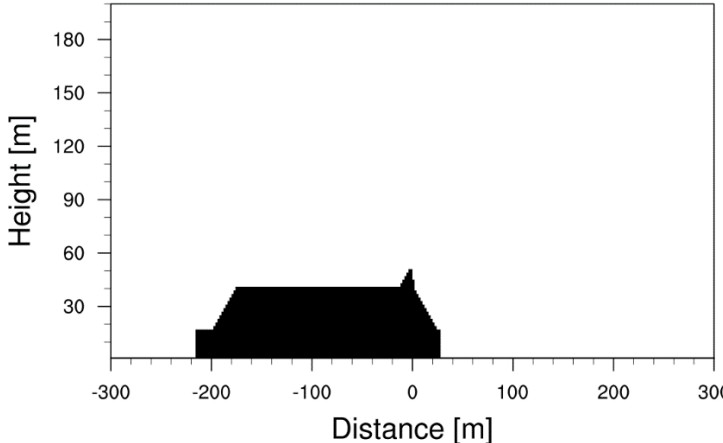

**Figure 3: Side view of the prototype cruise ship in the MITRAS domain with the x-axis located at the stack position.**



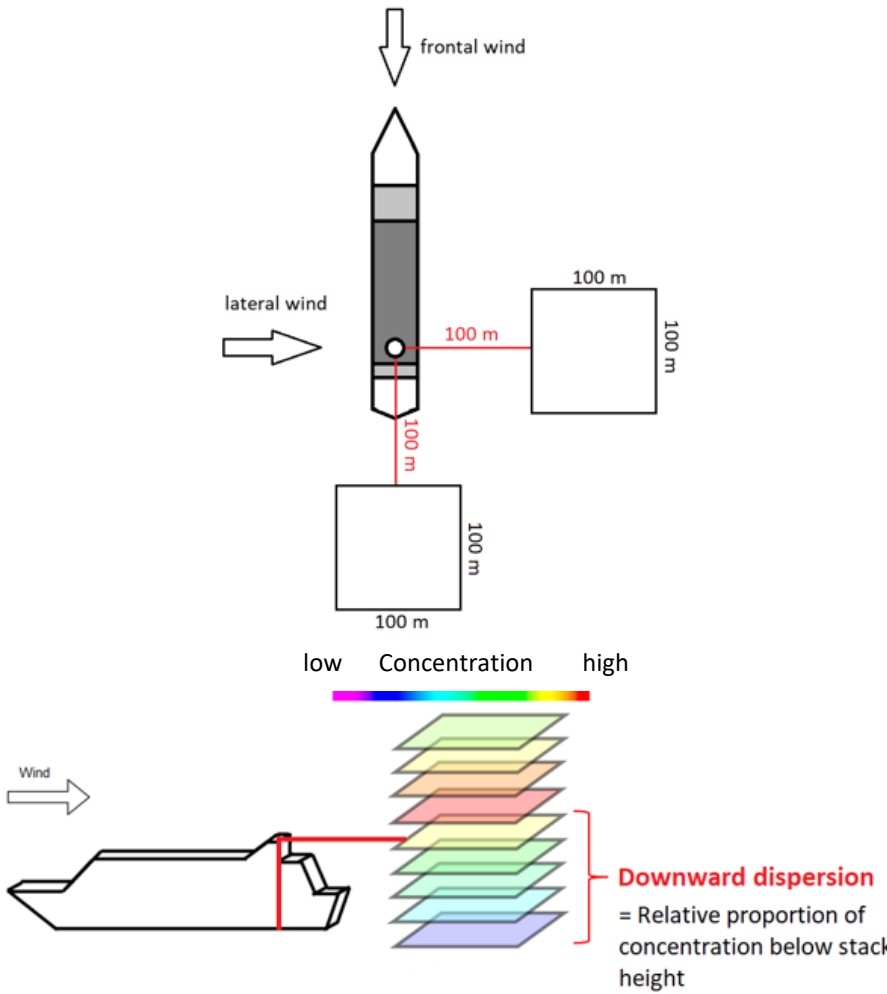


**Figure 4: Schematic sketches of the investigation area. Vertical concentration profiles are evaluated at a distance of 100 m downwind of the ship for layers of 100 m x 100 m.**





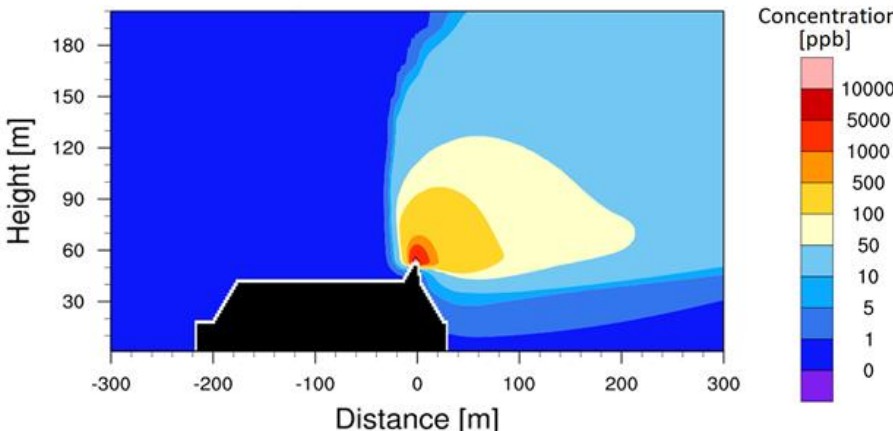

**Figure 5: MITRAS model results for default conditions (frontal wind at 5 m s⁻¹, exit velocity 10 m s⁻¹, exhaust temperature of 300 °C and an ambient temperature gradient of -0.65 K · 100 m⁻¹).**

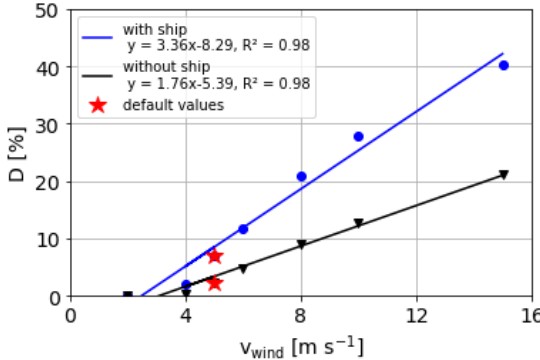

**Figure 6: Dependence of the downward dispersion D on different wind speeds v_wind with and without the obstacle effect.**

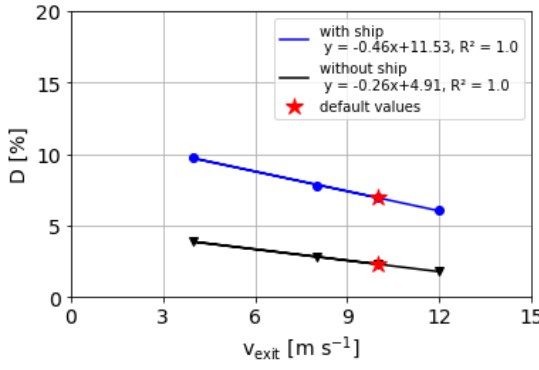

**Figure 7: Dependence of the downward dispersion D on different exit velocities v_exit with and without the obstacle effect.**





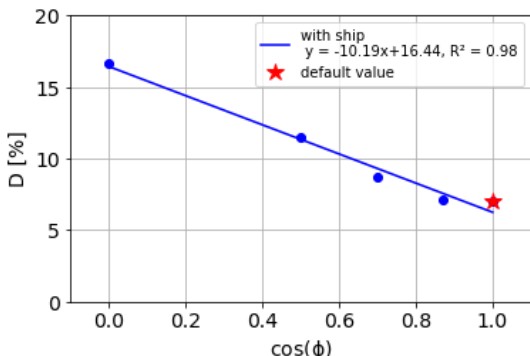

**Figure 8: Dependence of the downward dispersion D on the cosine of different wind flow angles (φ) towards the ship. 0° = frontal wind, 90° = lateral wind.**


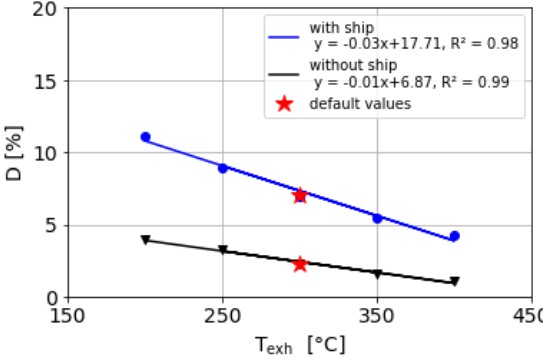

**Figure 9: Dependence of the downward dispersion D on different exhaust temperatures T$_{exh}$ with and without the obstacle effect.**





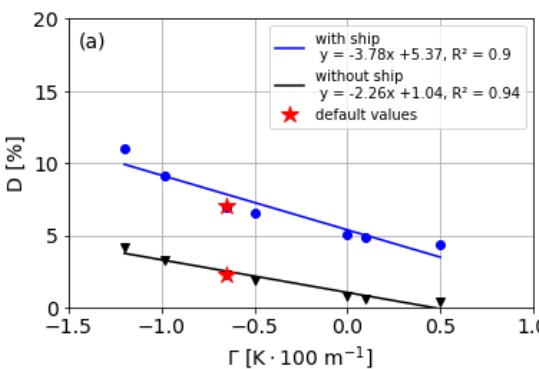


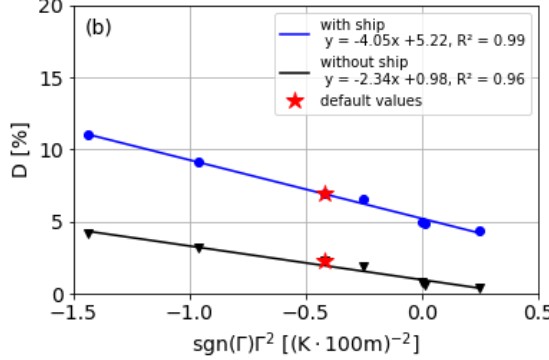

**Figure 10: Dependence of the downward dispersion D on different vertical temperature gradients Γ with and without the obstacle effect. Linear regressions of the downward dispersion against Γ (panel a) and sgn(Γ)Γ² (panel b) are shown.**




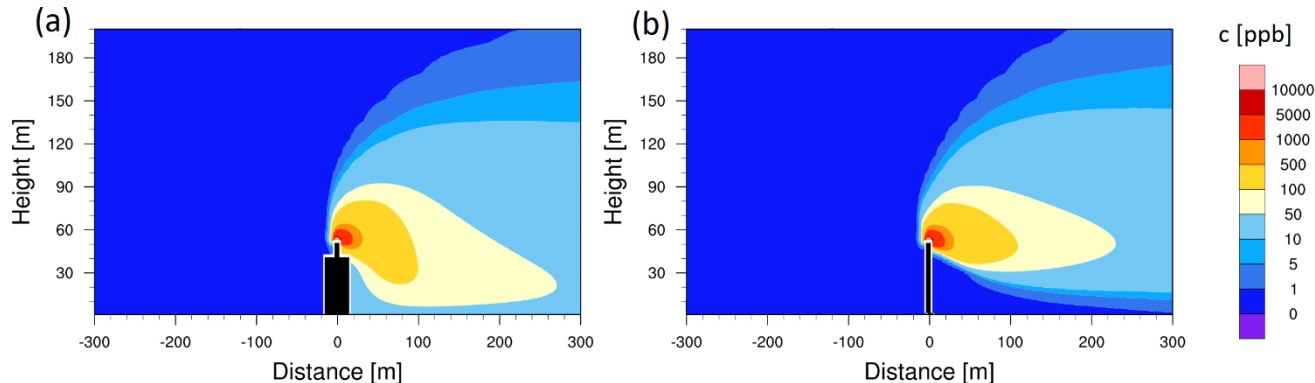

**Figure 11: Visualization of the obstacle effect in MITRAS. Examples for lateral wind with $v_{wind}$ = 15 m s$^{-1}$, $v_{exit}$ = 4 m s$^{-1}$, $T_{exh}$ = 200 °C and $\Gamma$ = -1.2 K · 100 m$^{-1}$ are presented. Concentration fields are displayed with ship, pointing towards the viewer (a) and under stack-only conditions (b).**


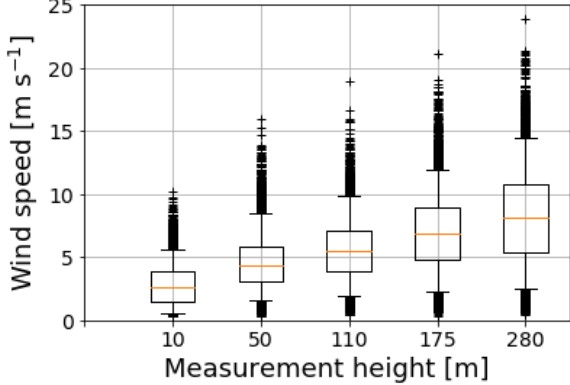

**Figure A1: Boxplots of hourly wind speed data for the year 2018 from Hamburg weather mast. Red lines indicate median values, lower and upper whiskers end at 5th and 95th percentile, respectively.**



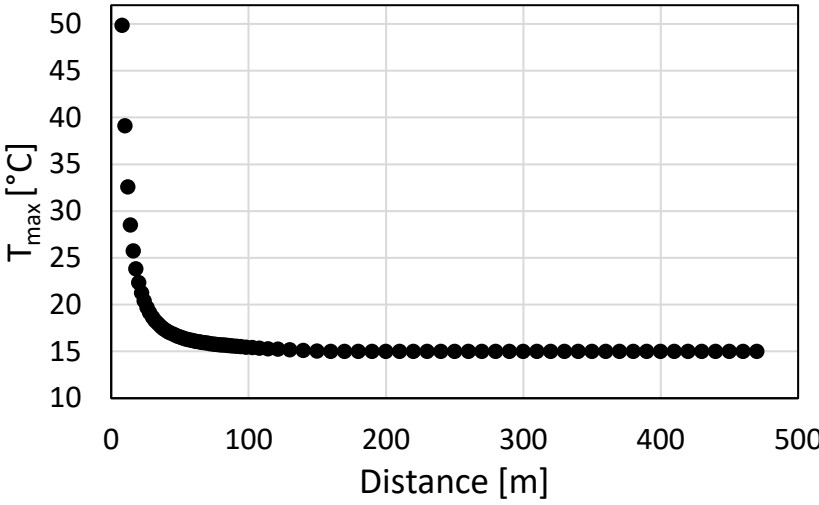


**Figure B1: Results for calculated maximum temperatures in the MITRAS domain in distance downwind from the stack for a case of 400 °C plume with ambient temperature of 15 °C. The values closer to the stack are not shown for clarity. The thermic effect on the plume rise due to the hot exhaust gas vanishes at around 100 m distance from the stack.**

**Tables**

**Table 1: Input parameters for this study. While varying one single input parameter in the investigation range, all others remain at default setting**

| Input Parameter | Default Setting | Investigation Range |
|---|---|---|
| Ambient temperature at surface | 15 °C | None |
| Ambient temperature gradient | -0.65 K · 100 m$^{-1}$ | -1.2–0.5 K · 100 m$^{-1}$ |
| Wind speed at upper model boundary | 5 m s$^{-1}$ | 2–15 m s$^{-1}$ |
| Wind direction | 0° (= frontal wind) | 0–90° |
| Ship length | 246 m | None |
| Ship width | 30 m | None |
| Stack height | 52 m | None |
| Exit velocity | 10 m s$^{-1}$ | 4–12 m s$^{-1}$ |
| Exhaust temperature | 300 °C | 200–400 °C |




**Table 2: Effective ranges of investigated input parameters on the downward dispersion under default settings**

| Input Parameter | Investigated Range | Default Value | Effective Range with Ship | Effective Range without Ship |
|---|---|---|---|---|
| Wind speed | 2–15 m s⁻¹ | 5 m s⁻¹ | 40.3 % | 21.1 % |
| Exit velocity | 4–12 m s⁻¹ | 10 m s⁻¹ | 3.7 % | 2.1 % |
| Wind direction | 0–90° | 0° (frontal) | 9.7 % | None |
| Exhaust temperature | 200–400 °C | 300 °C | 6.9 % | 2.9 % |
| Atmospheric stability | -1.2 to 0.5 K · 100 m⁻¹ | -0.65 K · 100 m⁻¹ | 6.6 % | 3.8 % |

**Table 3: Results of the bootstrapping procedure for cases with and without considering the ship-induced obstacle effect.**

| | With Ship | Without Ship |
|---|---|---|
| Number of training cases | 39 | 27 |
| Mean absolute error | 1.9 % | 1.2 % |
| Standard deviation | 1.6 % | 0.9 % |
| Maximum absolute error | 6.1 % | 4.0 % |


**Table A1: Statistical data on hourly wind speed values [m s⁻¹] from Hamburg weather mast for the year 2018.**

| | 10 m | 50 m | 110 m | 175 m | 280 m |
|---|---|---|---|---|---|
| Mean | 2.8 | 4.6 | 5.6 | 7.0 | 8.2 |
| Median | 2.6 | 4.3 | 5.5 | 6.9 | 8.1 |
| 5th percentile | 0.6 | 1.6 | 1.9 | 2.3 | 2.5 |
| 90th percentile | 5.0 | 7.4 | 8.7 | 10.8 | 13.1 |
| 95th percentile | 5.7 | 8.5 | 9.9 | 12.0 | 14.4 |
| 99th percentile | 7.2 | 10.8 | 12.1 | 14.6 | 16.9 |
| Maximum | 10.2 | 16.0 | 18.9 | 21.1 | 23.9 |



**Table C1: Data table for regression analyses. $v_{wind, in}$ refers to the input wind speed at the top model layer, $v_{wind, stack}$ refers to the wind speed at stack height. Further input data are exit velocity ($v_{exit}$), exhaust temperature ($T_{exh}$), wind direction ($\varphi$) with 0° referring to frontal and 90° to lateral wind. Results are given for downward dispersion with and without obstacle effect (D and $D_{stack-only}$). $D_{par}$ refers to results of the parameterization. The bold values in line number 8 corresponds to the default settings. Values in brackets in $D_{par, stack-only}$ were not included in the multiple regression, because in these cases only the wind direction was changed which does not**
**affect stack-only results.**

| # | $v_{wind, in}$ [m s$^{-1}$] | $v_{wind, stack}$ [m s$^{-1}$] | $v_{exit}$ [m s$^{-1}$] | $T_{exh}$ [°C] | $\varphi$ [°] | $\Gamma$ [K · 100 m$^{-1}$] | D [%] | $D_{stack-only}$ [%] | $D_{par}$ [%] | $D_{par, stack-only}$ [%] |
|---|---|---|---|---|---|---|---|---|---|---|
| 1 | 2.0 | 2.0 | 10.0 | 200 | 0 | -0.65 | 0.0 | 0.0 | 0.0 | -0.4 |
| 2 | 2.0 | 2.0 | 10.0 | 300 | 0 | -0.65 | 0.0 | 0.0 | -2.6 | -2.2 |
| 3 | 2.0 | 2.0 | 10.0 | 400 | 0 | -0.65 | 0.0 | 0.0 | -5.2 | -4.0 |
| 4 | 2.0 | 2.0 | 10.0 | 200 | 90 | -0.65 | 1.0 | 0.0 | 6.2 | (-0.4) |
| 5 | 2.0 | 2.1 | 10.0 | 300 | 90 | -0.65 | 0.7 | 0.0 | 3.6 | (-2.2) |
| 6 | 2.0 | 2.1 | 10.0 | 400 | 90 | -0.65 | 0.6 | 0.0 | 0.9 | (-4.0) |
| 7 | 5.0 | 4.7 | 10.0 | 200 | 0 | -0.65 | 11.2 | 3.9 | 10.4 | 4.9 |
| **8** | **5.0** | **4.7** | **10.0** | **300** | **0** | **-0.65** | **7.0** | **2.3** | **7.7** | **3.2** |
| 9 | 5.0 | 4.7 | 10.0 | 400 | 0 | -0.65 | 4.3 | 1.1 | 5.1 | 1.4 |
| 10 | 5.0 | 4.8 | 10.0 | 200 | 90 | -0.65 | 19.1 | 3.7 | 16.5 | (4.9) |
| 11 | 5.0 | 4.8 | 10.0 | 300 | 90 | -0.65 | 16.6 | 3.1 | 13.9 | (-2.2) |
| 12 | 5.0 | 4.8 | 10.0 | 400 | 90 | -0.65 | 13.6 | 2.1 | 11.3 | (1.4) |
| 13 | 8.0 | 6.9 | 4.0 | 200 | 0 | -0.65 | 32.1 | 14.8 | 26.8 | 14.1 |
| 14 | 8.0 | 6.9 | 4.0 | 300 | 0 | -0.65 | 24.7 | 10.8 | 24.2 | 12.3 |
| 15 | 8.0 | 6.9 | 4.0 | 400 | 0 | -0.65 | 19.7 | 8.6 | 21.6 | 10.5 |
| 16 | 8.0 | 7.3 | 4.0 | 200 | 90 | -0.65 | 34.6 | 16.9 | 32.9 | (14.1) |
| 17 | 8.0 | 7.3 | 4.0 | 300 | 90 | -0.65 | 30.8 | 12.4 | 30.3 | (12.3) |
| 18 | 8.0 | 7.3 | 4.0 | 400 | 90 | -0.65 | 27.6 | 9.9 | 27.7 | (10.5) |
| 19 | 5.0 | 4.7 | 10.0 | 250 | 0 | -0.65 | 8.9 | 3.3 | 9.1 | 4.1 |
| 20 | 5.0 | 4.7 | 10.0 | 350 | 0 | -0.65 | 5.4 | 1.6 | 6.4 | 2.3 |
| 21 | 4.0 | 3.9 | 10.0 | 300 | 0 | -0.65 | 2.2 | 0.3 | 4.3 | 1.4 |
| 22 | 6.0 | 5.4 | 10.0 | 300 | 0 | -0.65 | 11.8 | 4.9 | 11.2 | 5.0 |
| 23 | 8.0 | 6.9 | 10.0 | 300 | 0 | -0.65 | 20.9 | 9.0 | 18.1 | 8.5 |
| 24 | 10.0 | 8.3 | 10.0 | 300 | 0 | -0.65 | 28.0 | 12.8 | 25.0 | 12.1 |
| 25 | 5.0 | 4.7 | 4.0 | 300 | 0 | -0.65 | 9.8 | 3.9 | 13.8 | 6.9 |
| 26 | 5.0 | 4.7 | 8.0 | 300 | 0 | -0.65 | 7.8 | 2.8 | 9.8 | 4.4 |
| 27 | 5.0 | 4.7 | 12.0 | 300 | 0 | -0.65 | 6.1 | 1.8 | 5.7 | 1.9 |
| 28 | 5.0 | 5.2 | 10.0 | 300 | 0 | 0.50 | 4.4 | 0.4 | 5.2 | 0.9 |
| 29 | 5.0 | 5.0 | 10.0 | 300 | 0 | 0.10 | 4.9 | 0.6 | 6.1 | 1.7 |





| | | | | | | | | | | |
|---|---|---|---|---|---|---|---|---|---|---|
| 30 | 5.0 | 5.0 | 10.0 | 300 | 0 | 0.00 | 5.0 | 0.8 | 6.1 | 1.7 |
| 31 | 5.0 | 4.7 | 10.0 | 300 | 0 | -0.50 | 6.5 | 1.9 | 7.1 | 2.6 |
| 32 | 5.0 | 4.7 | 10.0 | 300 | 0 | -0.98 | 9.2 | 3.2 | 9.8 | 5.0 |
| 33 | 5.0 | 4.9 | 10.0 | 300 | 0 | -1.20 | 11.0 | 4.2 | 11.6 | 6.6 |
| 34 | 10.0 | 9.2 | 4.0 | 200 | 90 | -0.98 | 44.8 | 20.8 | 41.9 | 19.5 |
| 35 | 15.0 | 11.8 | 10.0 | 300 | 0 | -0.65 | 40.3 | 21.1 | 42.2 | 21.0 |
| 36 | 15.0 | 13.3 | 4.0 | 200 | 90 | -1.20 | 54.9 | 31.1 | 60.9 | 30.0 |
| 37 | 5.0 | 4.7 | 10.0 | 300 | 45 | -0.65 | 8.7 | 2.3 | 9.6 | (3.2) |
| 38 | 5.0 | 4.7 | 10.0 | 300 | 60 | -0.65 | 11.5 | 2.3 | 10.8 | (3.2) |
| 39 | 5.0 | 4.7 | 10.0 | 300 | 30 | -0.65 | 7.1 | 2.3 | 8.5 | (3.2) |