# Peer review of "Parameterizing the vertical downward dispersion of ship exhaust gas in the near-field"

_Atmospheric Chemistry and Physics, 2020_

## Referee Comment (RC1) · Anonymous Referee #1 · 19 Sep 2020

The paper deals with the simulation of the initial plume rise and dispersion of gas released by ship funnels. The topic is of big interest in the assessment of the impact of ship emissions in ports. The paper is well structured and clear.

I have some general comments that require a revision of the paper.

1) The paper can be defined as a study of the "building downwash" effect due to the ship itself. In this sense the paper is not really innovative, but results are interesting for the many different input parameters studied (wind velocity and direction, gas velocity and temperature and atmospheric stability class). Building downwash effect is present as option in many common dispersion models. Did the authors verified if the use of MITRAS gives different results with respect to, as an example, CALPUFF + building downwash?

2) Since the calculation domain is not very large a CFD model could be proposed to perform the same simulations. Do they authors think it could be more accurate?

3) Gas dispersion in the atmospheric boundary layer is a high non-linear process due to the effect of turbulence. But all the correlations reported in Fig.s 6-10 are well described by a simple linear equation between the downward dispersion D and all the input variables studied including also the temperature vertical gradients. Do the authors have an explanation for the very good linear correlation?

---

## Author Comment (AC1) · 26 Oct 2020

We appreciate the reviewer's opinion about the paper's topic and structure. Thank you very much for the comments.

C1: The paper can be defined as a study of the "building downwash" effect due to the ship itself. In this sense the paper is not really innovative, but results are interesting for the many different input parameters studied (wind velocity and direction, gas velocity and temperature and atmospheric stability class). Building downwash effect is present as option in many common dispersion models. Did the authors verified if the use of MITRAS gives different results with respect to, as an example, CALPUFF + building downwash?

Answer to C1:

As to our knowledge, CALPUFF is a Lagrangian Gaussian puff model that calculates building downwash effects via the Huber-Snyder or the Schulman-Scire model approach. Although it can calculate wind-direction in 36 times 10° increments, it only uses the relation of building width to height to account for the downward dispersion. Mitras is able to represent the complex building geometry of a ship with several decks. Therefore, the comparability is limited and we did not compare our results with CALPUFF. However, we did compare stack-only conditions with the integral plume model IBJPluris (Janicke and Janicke, 2001), which can be used to describe the plume dispersion in the momentum-driven regime. It calculates average plume properties like concentration and temperature along the plume centerline and applies a circular Gaussian dispersion around this central axis. IBJPluris does not account for obstacle-induced turbulence effects and is therefore only compared to stack-only conditions in MITRAS. Since the primary output of IBJPluris is the plume centerline and not the downward dispersion, we calculated a similar centerline height for MITRAS to compare the plume behavior. We define this centerline in MITRAS $h_{center, MITRAS}$ as the median height of the plume mass (i.e. 50 % of the plume mass lies below and 50 % lies above). It is calculated at the same distance as downward dispersion for a column of 100 m x 100 m (see Fig. 4). Since this is an average of values between a distance of 100 m to 200 m, we calculated IBJPluris centerline heights at a distance of 100 m to 200 m as well ($h_{center, Pluris}$). Table S1 gives an overview of the comparison. $\Delta h_{MITRAS}$ and $\Delta h_{Pluris}$ are the differences between plume height (52 m) and centerline height for MITRAS and IBJPluris calculations, respectively. Their minimum difference is given by $\min(|\Delta h_{MITRAS} - \Delta h_{Pluris}|)$ in Table S1, which represents the closest similarity of both models. Results are given at default settings and selected conditions to compare effects of input parameters. For all selected cases, MITRAS calculates larger centerline height values than IBJPluris. The lowest differences occur at low wind speed, low exhaust temperature and very stable conditions. The strongest differences of over 20 m occur for cases of low exit velocity and high exhaust temperature. By calculating

effective ranges,

ri = Δ|min(|ΔhMITRAS - ΔhPluris|)i, max - min(|ΔhMITRAS - ΔhPluris|)i, min|

for a certain input parameter i, one can evaluate which input parameter causes the highest discrepancy between the models. For example, changing the wind speed only results in an effective range of 1 m, while temperature and stability changes both show effective ranges of 10 m. The higher plume rise in MITRAS is consistent with the interaction of the hot plume with the ambient air. MITRAS accounts for the change in the thermodynamic field and the heat balance equation creates an additional buoyancy which is not considered in the simpler approaches. This explains the high effective range for temperature and stability changes. We conclude that the results for stack-only conditions are reasonable and that MITRAS provides a more complex improvement over simple Gaussian approaches. We will include this comparison in the appendix of the paper.

C2: Since the calculation domain is not very large a CFD model could be proposed to perform the same simulations. Do they authors think it could be more accurate?

Answer to C2:

As an obstacle-resolving microscale meteorology model MITRAS can be understood as a CFD model for the atmosphere, but with additional processes taken into account such as Coriolis effect and precipitation. In this special application, effects of the Coriolis force are considered. If necessary, it is possible to create obstacles at even higher resolution. MITRAS has been evaluated following an evaluation protocol that includes the comparison of wind fields for different obstacle geometries with wind tunnel data (Grawe et al., 2013).

C3: For most of the input parameters, the correlation is directly linear. However, in case of atmospheric stability the downward dispersion is linear against a derived parameter $sgn(\Gamma)\Gamma^2$, which has to be distinguished from a direct linear correlation. The good correlations result from the way this downward dispersion is calculated. Due to calculating a column average (above or beyond the stack height) of a column with 100 m x 100 m surface area, a lot of the mentioned turbulence is averaged out. If one calculates the downward dispersion at one specific grid cell, the correlation would probably be lower. However, the averaged results are better to be coupled into city-scale models which is why we preferred them over single-grid cell results.

Answer to C3:

For most of the input parameters, the correlation is directly linear. However, in case of atmospheric stability the downward dispersion is linear against a derived parameter $\mathrm{sgn}(\Gamma)\Gamma^2$, which has to be distinguished from a direct linear correlation. The good correlations result from the way this downward dispersion is calculated. Due to calculating a column average (above or beyond the stack height) of a column with 100 m x 100 m surface area, a lot of the mentioned turbulence is averaged out. If one calculates the downward dispersion at one specific grid cell, the correlation would probably be lower. However, the averaged results are better to be coupled into city-scale models which is why we preferred them over single-grid cell results.

Please also note the supplement to this comment:
https://acp.copernicus.org/preprints/acp-2020-753/acp-2020-753-AC1-supplement.pdf

[Figure]

**Supplement:**

**Table S1:** Comparison of results of plume center heights $h_{center}$ for MITRAS and IBJPluris for different selected cases (case numbers based on Table C1).

| Case name & number | $v_{wind}$ [m s$^{-1}$] | $v_{exit}$ [m s$^{-1}$] | $T_{exh}$ [°C] | $\Gamma$ [K · 100 m$^{-1}$] | MITRAS | | IBJPluris | | min(\|$\Delta h_{MITRAS}$ - $\Delta h_{Pluris}$\|) [m] |
|---|---|---|---|---|---|---|---|---|---|
| | | | | | $h_{center, MITRAS}$ [m] | $\Delta h_{MITRAS}$ [m] | $h_{center, Pluris}$ [m] | $\Delta h_{Pluris}$ [m] | |
| Default settings (8) | 5 | 10 | 300 | -0.65 | 110 | 58 | 76 – 92 | 24 – 40 | 18 |
| Low wind speed (2) | 2 | 10 | 300 | -0.65 | 160 | 108 | 117 – 151 | 65 – 99 | 9 |
| high wind speed (36) | 15 | 10 | 300 | -0.65 | 76 | 24 | 61 – 66 | 9 – 14 | 10 |
| low exit velocity (25) | 5 | 4 | 300 | -0.65 | 104 | 52 | 71 – 82 | 19 – 30 | 22 |
| high exit velocity (27) | 5 | 12 | 300 | -0.65 | 112 | 60 | 79 – 95 | 27 – 43 | 17 |
| low exhaust temperature (7) | 5 | 10 | 200 | -0.65 | 102 | 50 | 77 – 91 | 25 – 39 | 11 |
| high exhaust temperature (9) | 5 | 10 | 400 | -0.65 | 116 | 64 | 79 – 95 | 27 – 43 | 21 |
| unstable condition (33) | 5 | 10 | 300 | -1.2 | 112 | 60 | 78 – 93 | 26 – 41 | 19 |
| very stable condition (28) | 5 | 10 | 300 | +0.5 | 100 | 48 | 77 – 91 | 25 – 39 | 9 |

---

## Referee Comment (RC2) · Anonymous Referee #2 · 29 Jan 2021

This is an interesting paper on the vertical downward dispersion of a prototype ship exhaust gas in the near-field and its dependence upon several input parameters, using MITRAS model simulations. The topic is actual and important. The paper does not present substantial new concepts. However, the considerable number of combinations of input parameters makes the paper of great interest. The paper contains a significant number of assumptions, but this is impossible to avoid given the complexity of the case under consideration. The paper is well-structured, the results are discussed properly.

I have some comments which would improve the quality of the paper but are not essential for publication.

1) I would like to see a bit more about the plume rise calculation and the parameterization of entrainment coefficient in MITRAS. Please give one formula for better under-

standing the physical processes involved in plume rise. This may be helpful for those readers who are not familiar with it.

2) As you mentioned, several authors pointed out that Gaussian pollutant distribution might not be well suited in the near field. When the distribution is asymmetric, the perfect reflection from the surface may not be the best choice. Did the authors consider using a more sophisticated algorithm for reflection?

3)Two multiple regressions are performed (with and without ship). It would be interesting to vary the prototype shape and include it as independent variable (e.g., aspect ratio, length/width).

4) The emission is assumed to occur in grid cell (2m x 2m x 2m), but the real stack is usually round and have a smaller diameter. This is an intrinsic problem of Eulerian modelling. Did the authors consider comparing their results with other dispersion models (e.g., Lagrangian particles models)?

———————————————————

---

## Author Comment (AC2) · 18 Feb 2021

We appreciate the reviewer's positive feedback on the paper. Thank you very much for the comments.

C1: I would like to see a bit more about the plume rise calculation and the parameterization of entrainment coefficient in MITRAS. Please give one formula for better understanding the physical processes involved in plume rise. This may be helpful for those readers who are not familiar with it.

Answer to C1:

We did not quantify plume rise itself, but the downward dispersion that differs depending on the input parameters that affect vertical movement like exhaust temperature,

exit velocity, wind speed and stability. The prognostic model MITRAS calculates concentration changes based on the Navier-Stokes Equations, continuity equation and conservation equations for scalar properties on an Eulerian grid. Therefore, entrainment is not parameterized in MITRAS, but is rather a result of turbulence caused by the ship and the exhaust temperature. For turbulent kinetic energy and vertical exchange coefficients, we applied the Prandtl-Kolmogorov closure (Chapter 3.4.5 in Schlünzen et al. 2018). Furthermore, advection and diffusion terms in the Reynolds equations are integrated in time by the means of the Adams-Bashfort-scheme (Chapter B.1.1 in Schlünzen et al., 2018).

Reference: Schlünzen, K. H., Boettcher, M., Fock, B. H., Gierisch, A., Grawe, D. and Salim, M.: Scientific Documentation of the Multiscale Model System M-SYS, MEMI Tech. Rep. 4, CEN, Univ. Hambg., 1–153, 2018.

This reference will also replace the Schluenzen et al. (2012) reference in the previous paper version as a more recent reference of the modelling system.

C2: As you mentioned, several authors pointed out that Gaussian pollutant distribution might not be well suited in the near field. When the distribution is asymmetric, the perfect reflection from the surface may not be the best choice. Did the authors consider using a more sophisticated algorithm for reflection?

Answer to C2:

The Eulerian microscale model MITRAS does not assume reflection on the ground, but calculates physical processes like advection and dispersion inside the Eulerian grid volume. It can therefore be considered a more sophisticated model to account for near-surface processes. We will cover the parameterization of the vertical plume profile and the possible uncertainties of a Gaussian profile for vertical plume concentration distribution in a future study.

C3: Two multiple regressions are performed (with and without ship). It would be interesting to vary the prototype shape and include it as independent variable (e.g., aspect ratio, length/width)

Answer to C3:

Yes, this is an interesting suggestion and may be included in a future study. There are numerous factors in the ship geometry that can affect the plume movement (additionally to your mentioned parameters also the location of the stack, the proportion of the vessel above sea surface and the stack height). Including all of these factors may give additional interesting insights, but is beyond the scope of the parameterization in this study and may be solved better by a machine-learning algorithm. As mentioned, we chose the prototype size based on the average cruise ship traffic in Hamburg to investigate the average effect of a cruise-ship sized vessel on pollutant concentrations close to ground. To get an impression, we added an exemplary comparison of MITRAS results for a cruise ship and container vessel in Supplementary S2. The size of the ship has a significant impact on the downward dispersion. For the smaller container vessel, we found a downward dispersion of only 33 % compared to 43 % for the cruise ship for cold plume conditions.

C4: The emission is assumed to occur in grid cell (2m x 2m x 2m), but the real stack is usually round and have a smaller diameter. This is an intrinsic problem of Eulerian modelling. Did the authors consider comparing their results with other dispersion models (e.g., Lagrangian particles models)?

Answer to C4:

Reviewer 1 has raised a similar question. We compared MITRAS results with the integral plume model IBJPluris (Janicke and Janicke 2001). See out response to question 1 by reviewer 1 for a comparison of MITRAS with IBJPluris. We will also include this comparison in the final version of the paper. Regarding the diameter of the stack, it is true that the stack diameter in our study lies in the upper range of real ship stack diameters. However, Bai et al. (2020) reported about plume modeling for container vessels

with measured funnel diameters in the range of 1.38 to 3.0 m. Furthermore, many ships operate multiple smaller stacks that might in sum lead to a similar exhaust behavior. In general, reducing the stack diameter at the same exhaust flow and temperature leads to a reduced plume rise and an increased downward dispersion.

Reference: Bai, S., Wen, Y., He, L., Liu, Y., Zhang, Y., Yu, Q. and Ma, W.: Single-Vessel Plume Dispersion Simulation: Method and a Case Study Using CALPUFF in the Yantian Port Area, Shenzhen (China), Int. J. Environ. Res. Public Health, 1–29, doi:10.3390/ijerph17217831, 2020.

Please also note the supplement to this comment:
https://acp.copernicus.org/preprints/acp-2020-753/acp-2020-753-AC2-supplement.pdf

———————————————————

[Figure]

**Supplement:**

**Table S2:** Exemplary comparison of MITRAS results for two different ship types.

| Inputs | Cruise Ship | Container Ship |
|---|---|---|
| Length | 246 m | 168 m |
| Width | 30 m | 27 m |
| Height stack | 52 m | 38 m |
| Wind speed at stack height | 5 ms$^{-1}$ | |
| Wind direction | frontal | |
| Exit velocity | 10 ms$^{-1}$ | |
| Exhaust temperature | 15°C | |
| Stability | -0.65 K · 100 m$^{-1}$ | |
| | | |
| **Output**
 Downward Dispersion in 100 m distance | 43 % | 33 % |

---

## Author Response (AR1)

**Comments from Referee 1 (R1)**

The paper deals with the simulation of the initial plume rise and dispersion of gas released by ship funnels. The topic is of big interest in the assessment of the impact of ship emissions in ports. The paper is well structured and clear.

**Authors response:**

We appreciate the reviewer's opinion about the paper's topic and structure.

**R1C1:**

I have some general comments that require a revision of the paper.

The paper can be defined as a study of the "building downwash" effect due to the ship itself. In this sense the paper is not really innovative, but results are interesting for the many different input parameters studied (wind velocity and direction, gas velocity and temperature and atmospheric stability class). Building downwash effect is present as option in many common dispersion models. Did the authors verified if the use of MITRAS gives different results with respect to, as an example, CALPUFF + building downwash? C1

**Author's response to R1C1:**

Thank you for the comment.

As to our knowledge, CALPUFF is a Lagrangian Gaussian puff model that calculates building downwash effects via the Huber-Snyder or the Schulman-Scire model approach. Although it can calculate wind-direction in 36 times 10° increments, it only uses the relation of building width to height to account for the downward dispersion. Mitras is able to represent the complex building geometry of a ship with several decks. Therefore, the comparability is limited and we did not compare our results with CALPUFF.

However, we did compare stack-only conditions with the integral plume model IBJPluris (Janicke and Janicke, 2001), which can be used to describe the plume dispersion in the momentum-driven regime. It calculates average plume properties like concentration and temperature along the plume centerline and applies a circular Gaussian dispersion around this central axis. IBJPluris does not account for obstacle-induced turbulence effects and is therefore only compared to stack-only conditions in MITRAS.

Since the primary output of IBJPluris is the plume centerline and not the downward dispersion, we calculated a similar centerline height for MITRAS to compare the plume behavior. We define this centerline in MITRAS $h_{center, MITRAS}$ as the median height of the plume mass (i.e. 50 % of the plume mass lies below and 50 % lies above). It is calculated at the same distance as downward dispersion for a column of 100 m x 100 m (see Fig. 4). Since this is an average of values between a distance of 100 m to 200 m, we calculated IBJPluris centerline heights at a distance of 100 m to 200 m as well ($h_{center, Pluris}$). Table S1 gives an overview of the comparison. $\Delta h_{MITRAS}$ and $\Delta h_{Pluris}$ are the differences between plume height (52 m) and centerline height for MITRAS and IBJPluris calculations, respectively. Their minimum difference is given by $min(|\Delta h_{MITRAS} - \Delta h_{Pluris}|)$ in Table S1, which represents the closest similarity of both models. Results are given at default settings and selected conditions to compare effects of input parameters.

For all selected cases, MITRAS calculates larger centerline height values than IBJPluris. The lowest differences occur at low wind speed, low exhaust temperature and very stable conditions.

The strongest differences of over 20 m occur for cases of low exit velocity and high exhaust temperature.

By calculating effective ranges,

$$r_i = \Delta|\min(|\Delta h_{MITRAS} - \Delta h_{Pluris}|)_{i,\,max} - \min(|\Delta h_{MITRAS} - \Delta h_{Pluris}|)_{i,\,min}|$$

for a certain input parameter i, one can evaluate which input parameter causes the highest discrepancy between the models. For example, changing the wind speed only results in an effective range of 1 m, while temperature and stability changes both show effective ranges of 10 m.

The higher plume rise in MITRAS is consistent with the interaction of the hot plume with the ambient air. MITRAS accounts for the change in the thermodynamic field and the heat balance equation creates an additional buoyancy which is not considered in the simpler approaches. This explains the high effective range for temperature and stability changes.

We conclude that the results for stack-only conditions are reasonable and that MITRAS provides a more complex improvement over simple Gaussian approaches.

**Author's changes in the manuscript according to R1C1:**
*Line specifications correspond to the lines in the track-change file.

- The additional appendix has been mentioned in the main text lines 265-266
- The comparison with IBJPluris has been added as appendix D, line 487-513
- Table S1 has been added to the Supplementary

**R1C2:**
Since the calculation domain is not very large a CFD model could be proposed to perform the same simulations. Do they authors think it could be more accurate?

**Author's response to R1C2:**
As an obstacle-resolving microscale meteorology model MITRAS can be understood as a CFD model for the atmosphere, but with additional processes taken into account such as Coriolis effect and precipitation. In this special application, effects of the Coriolis force are considered. If necessary, it is possible to create obstacles at even higher resolution. MITRAS has been evaluated following an evaluation protocol that includes the comparison of wind fields for different obstacle geometries with wind tunnel data (Grawe et al., 2013).

**Author's changes in the manuscript according to R1C2:**
No changes were made in the manuscript.

**R1C3:**
Gas dispersion in the atmospheric boundary layer is a high non-linear process due to the effect of turbulence. But all the correlations reported in Figs. 6-10 are well de-scribed by a simple

linear equation between the downward dispersion D and all the input variables studied including also the temperature vertical gradients. Do the authors have an explanation for the very good linear correlation?

**Author's response to C3:**
For most of the input parameters, the correlation is directly linear. However, in case of atmospheric stability the downward dispersion is linear against a derived parameter $\text{sgn}(\Gamma)\Gamma^2$, which has to be distinguished from a direct linear correlation.

The good correlations result from the way this downward dispersion is calculated. Due to calculating a column average (above or beyond the stack height) of a column with 100 m x 100 m surface area, a lot of the mentioned turbulence is averaged out. If one calculates the downward dispersion at one specific grid cell, the correlation would probably be lower. However, the averaged results are better to be coupled into city-scale models which is why we preferred them over single-grid cell results.

**Author's changes in the manuscript according to R1C3:**
No changes were made in the manuscript.

**Comments from Referee 2 (R2):**
This is an interesting paper on the vertical downward dispersion of a prototype ship exhaust gas in the near-field and its dependence upon several input parameters, using MITRAS model simulations. The topic is actual and important. The paper does not present substantial new concepts. However, the considerable number of combinations of input parameters makes the paper of great interest. The paper contains a significant number of assumptions, but this is impossible to avoid given the complexity of the case under consideration. The paper is well-structured, the results are discussed properly. I have some comments which would improve the quality of the paper but are not essential for publication.

**Authors response:**
We appreciate the reviewer's positive feedback on the paper. Thank you very much for the comments.

**R2C1:**
I would like to see a bit more about the plume rise calculation and the parameterization of entrainment coefficient in MITRAS. Please give one formula for better understanding the physical processes involved in plume rise. This may be helpful for those readers who are not familiar with it.

**Author's response to R2C1:**
We did not quantify plume rise itself, but the downward dispersion that differs depending on the input parameters that affect vertical movement like exhaust temperature, exit velocity,

wind speed and stability. The prognostic model MITRAS calculates concentration changes based on the Navier-Stokes Equations, continuity equation and conservation equations for scalar properties on an Eulerian grid. Therefore, entrainment is not parameterized in MITRAS, but is rather a result of turbulence caused by the ship and the exhaust temperature.

For turbulent kinetic energy and vertical exchange coefficients, we applied the Prandtl-Kolmogorov closure (Chapter 3.4.5 in Schlünzen et al. 2018).

Furthermore, advection and diffusion terms in the Reynolds equations are integrated in time by the means of the Adams-Bashfort-scheme (Chapter B.1.1 in Schlünzen et al., 2018).

New reference:

Schlünzen, K. H., Boettcher, M., Fock, B. H., Gierisch, A., Grawe, D. and Salim, M.: Scientific Documentation of the Multiscale Model System M-SYS, MEMI Tech. Rep. 4, CEN, Univ. Hambg., 1–153, 2018.

**Author's changes in the manuscript according to R2C1:**
- The reference Schlünzen et al. (2012) has been replaced by the more recent version of the model description of Schlünzen et al. (2018)
- Corresponding lines: 89, 93 and 427-431 in the reference list

**R2C2:**
As you mentioned, several authors pointed out that Gaussian pollutant distribution might not be well suited in the near field. When the distribution is asymmetric, the perfect reflection from the surface may not be the best choice. Did the authors consider using a more sophisticated algorithm for reflection?

**Author's Response to R2C2:**
The Eulerian microscale model MITRAS does not assume reflection on the ground, but calculates physical processes like advection and dispersion inside the Eulerian grid volume. It can therefore be considered a more sophisticated model to account for near-surface processes.

We will cover the parameterization of the vertical plume profile and the possible uncertainties of a Gaussian profile for vertical plume concentration distribution in a future study.

**Author's changes in the manuscript according to R2C2:**
No changes were made in the manuscript.

**R2C3:**

Two multiple regressions are performed (with and without ship). It would be interesting to vary the prototype shape and include it as independent variable (e.g., aspect ratio, length/width)

**Author's response to R2C3:**

Yes, this is an interesting suggestion and may be included in a future study.

There are numerous factors in the ship geometry that can affect the plume movement (additionally to your mentioned parameters also the location of the stack, the proportion of the vessel above sea surface and the stack height). Including all of these factors may give additional interesting insights, but is beyond the scope of the parameterization in this study and may be solved better by a machine-learning algorithm. As mentioned, we chose the prototype size based on the average cruise ship traffic in Hamburg to investigate the average effect of a cruise-ship sized vessel on pollutant concentrations close to ground.

To get an impression, we added an exemplary comparison of MITRAS results for a cruise ship and container vessel in Supplementary S2. The size of the ship has a significant impact on the downward dispersion. For the smaller container vessel, we found a downward dispersion of only 33 % compared to 43 % for the cruise ship for cold plume conditions.

**Author's changes in the manuscript according to R2C3:**
- Additional information on the shape effect have been added at lines 290-293
- Table S2 has been added to the Supplementary

**R2C4:**

The emission is assumed to occur in grid cell (2m x 2m x 2m), but the real stack is usually round and have a smaller diameter. This is an intrinsic problem of Eulerian modelling. Did the authors consider comparing their results with other dispersion models (e.g., Lagrangian particles models)?

**Author's response to R2C4:**

Reviewer 1 raised a similar question. We compared MITRAS results with the integral plume model IBJPluris (Janicke and Janicke 2001). See out response to question 1 by reviewer 1 for a comparison of MITRAS with IBJPluris. We will also include this comparison in the final version of the paper.

Regarding the diameter of the stack, it is true that the stack diameter in our study lies in the upper range of real ship stack diameters. However, Bai et al. (2020) reported about plume modeling for container vessels with measured funnel diameters in the range of 1.38 to 3.0 m. Furthermore, many ships operate multiple smaller stacks that might in sum lead to a similar exhaust behavior.

In general, reducing the stack diameter at the same exhaust flow and temperature leads to a reduced plume rise and an increased downward dispersion.

New reference:

Bai, S., Wen, Y., He, L., Liu, Y., Zhang, Y., Yu, Q. and Ma, W.: Single-Vessel Plume Dispersion Simulation: Method and a Case Study Using CALPUFF in the Yantian Port Area, Shenzhen (China), Int. J. Environ. Res. Public Health, 1–29, doi:10.3390/ijerph17217831, 2020.

**Author's changes in the manuscript according to R2C4:**
- Information on stack diameters and the new source Bai et al. (2020) have been added at lines 280-283 and 342-344

**Other changes**
- A value has been corrected in table C1 (independent on referee comments)